# Water Quality Assessment and Environmental Impact of Heavy Metals in the Red Sea Coastal Seawater of Yanbu, Saudi Arabia

Abdelbaset S. El-Sorogy [ID], Mohamed Youssef *[ID] and Mansour H. Al-Hashim

Geology and Geophysics Department, College of Science, King Saud University, Riyadh 11451, Saudi Arabia
* Correspondence: mymohamed@ksu.edu.sa; Tel.: +966-556-545-897

**Abstract:** The Yanbu industrial city along the Red Sea coast includes industries associated with crude oil and natural gas production and refining and support industries that produce manufactured goods for domestic and/or internal consumption. This study investigates the potential environmental impact and the possible sources of heavy metals (HMs), and it evaluates the quality of coastal surface seawater in the vicinity of Yanbu, along the Red Sea coast of Saudi Arabia. Thirty seawater samples have been collected and analyzed using an inductively coupled plasma mass spectrometer (ICP-MS) in order to determine the concentration values of Fe, Cr, Pb, Sb, Mn, Cu, Zn, Al, Ni, As, Cd, Co, and Hg. Reported HMs averages (μg/L) are in the following sequence: Ni (4.424) > As (4.297) > Cu (2.447) > Zn (1.667) > Al (1.133) > Fe (0.983) > Cr (0.723) > Mn (0.328) > Cd (0.309) > Pb (0.276) > Sb (0.238) > Co (0.144) > Hg (0.058). The contamination index (Cd) showed low contamination levels in all of the analyzed samples, whereas the index of heavy metal pollution (HPI) revealed medium contamination levels in 28 samples and low levels in two samples. Reported high HMs variations within samples are attributed to the multiplication of sources. The statistical analyses indicated anthropogenic sources for Cd, Co, Hg, Zn, and Ni, which may have originated from industrial, farming, or fishing activities around Yanbu city, while the remaining metals might be originated from combined lithogenic and human sources.

**Keywords:** water quality; environmental impact evaluation; heavy metals; seawaters; Yanbu coast; Arabian Gulf

## 1. Introduction

Metals input into the environment may have a profound negative impact on several environmental components such as air, water, and soil [1–4]. Transitional environments such as coasts and beaches, which present where land and sea interact, are of particular interest due to their boundaries being unequivocally defined by their dynamics, shape, and functions [5–10]. Naturally occurring HMs are characterized by low concentration values. However, these natural levels have increased tremendously due to ever-increasing human activities [11,12]. HM contamination has become a serious environmental issue in coastal environments [13–17]. Increasing HM levels from natural and anthropogenic sources lead to environmental deterioration and consequent hazards due to bio-accumulation trophic transfer processes [18–21]. Although at normal levels, HMs are considered crucial for metabolism in living organisms and play an important role in plant growth, they become toxic at higher levels (e.g., Cu, Co, Zn, and Ni) [22]. However, some other HMs are considered highly toxic even when present at very low concentration levels (e.g., Hg, As, Pb, Cr, and Cd) [23,24].

The steady accumulation of HMs in coastal seawater from domestic, industrial, agricultural, tourism, fishing, and oil shipping and storage activities raises significant concerns and creates a societal health risk [25–27]. Due to this continuous accumulation of HMs in the aquatic ecosystem, toxic concentration levels are commonly detected in such environments, which pose serious risks to human health and to other living organisms alike [12].

Accordingly, assessment of HM contamination in seawater is one of the best mitigation steps for these potential risks to marine habitats and marine resources, and it is of great interest in ecotoxicology and environmental management [28–30].

The coastal environments of western Saudi Arabia have been subjected to intensive environmental studies using assessment of HMs in coastal sediments, e.g., [16,17,31–35]. Previous studies of coastal sediments in the Yanbu area have concluded that concentration levels of HMs range from very high risk (Cd, Hg Cu, Sb, and As) to moderate risk (Pb and Ni) and to low risk (Zn and Cr) [35]. Moreover, they attributed the anthropogenic contamination to agricultural, fishing, industrial, and urbanization activities. Natural HMs sources such as atmospheric inputs and rock weathering have also been implicated in these studies. Acute and likely chronic health effects have long been linked to water and food contaminated with high levels of HMs, including, for instance, blue-baby syndrome or methemoglobinemia [36]. Nonetheless, continuous monitoring of coastal Red Sea environments using seawater is very rare. Therefore, the current investigation uses multivariate statistical analyses and employs several pollution indices in an attempt to evaluate the degree of seawater pollution with HMs along the Yanbu coastal area, Saudi Arabia, and to pinpoint their potential origin.

## 2. Methods

### 2.1. Location and Sampling

Situated along the Red Sea coast of western Saudi Arabia, the study area extends between latitudes 23°40′33″–24°15′52″ N and longitudes 38°29′11″–37°43′03″ E (Figure 1). Sandy coastline characterizes the area, with only a few swamps, due to weathering of geological formations nearby the study area and the transportation of siliciclastic deposits toward the coastline [36]. Thirty seawater samples were collected from the shallow subtidal zone (Figure 1). Sampling locations were selected carefully along the Red Sea coast of Saudi Arabia either nearby the locations of human activities: agricultural, fishing, industrial, and urbanization activities. As a precautionary measure, 500 mL clean, sterilized plastic bottles were used for sample collection and were washed with seawater several times before sampling [6,10]. Acidification with 5 mL 10% $HNO_3$ was utilized in order to prevent precipitation and the sorption of water samples onto the bottles [37]. Measurements such as electrical conductivity, pH, and salinity were directly taken on site. The heavy metals in question (Fe, Mn, Cu, Cr, Pb, Sb, Zn, Al, Ni, As, Cd, Co, and Hg) were measured using inductively coupled plasma mass spectrometer (ICP-MS): NexION 300 D (Perkin Elmer, Waltham, MA, USA) at King Saud University. In the lab, and prior to analysis, deionized water was used to dilute the collected samples up to 10 times. Following filtration by a 0.45 μm micropore membrane filter, the samples were refrigerated at 4 °C, and then 50 mL of each sample was transferred to a beaker before being digested with 5 mL of concentrated $HNO_3$ to extract the metals. The ICP-MS was externally calibrated. Moreover, the measured concentrations of HMs in seawater samples were compared with the same HMs reported from coastal sediments of the Yanbu area. The calibration curves of the 6 elements: Cr, Cd, Cu, Pb, Co, and Ni, were obtained using the blank and three working standards 0, 50, 100, and 200 μg/L (Panreac, 766333. 1208). For As and Zn elements, we used the blank and three working standards 0, 50, 100, and 200 μg/L (Aristar grade, BDH laboratory supplies, England for the heavy metals). Calibration curves showed excellent linearity for all elements.

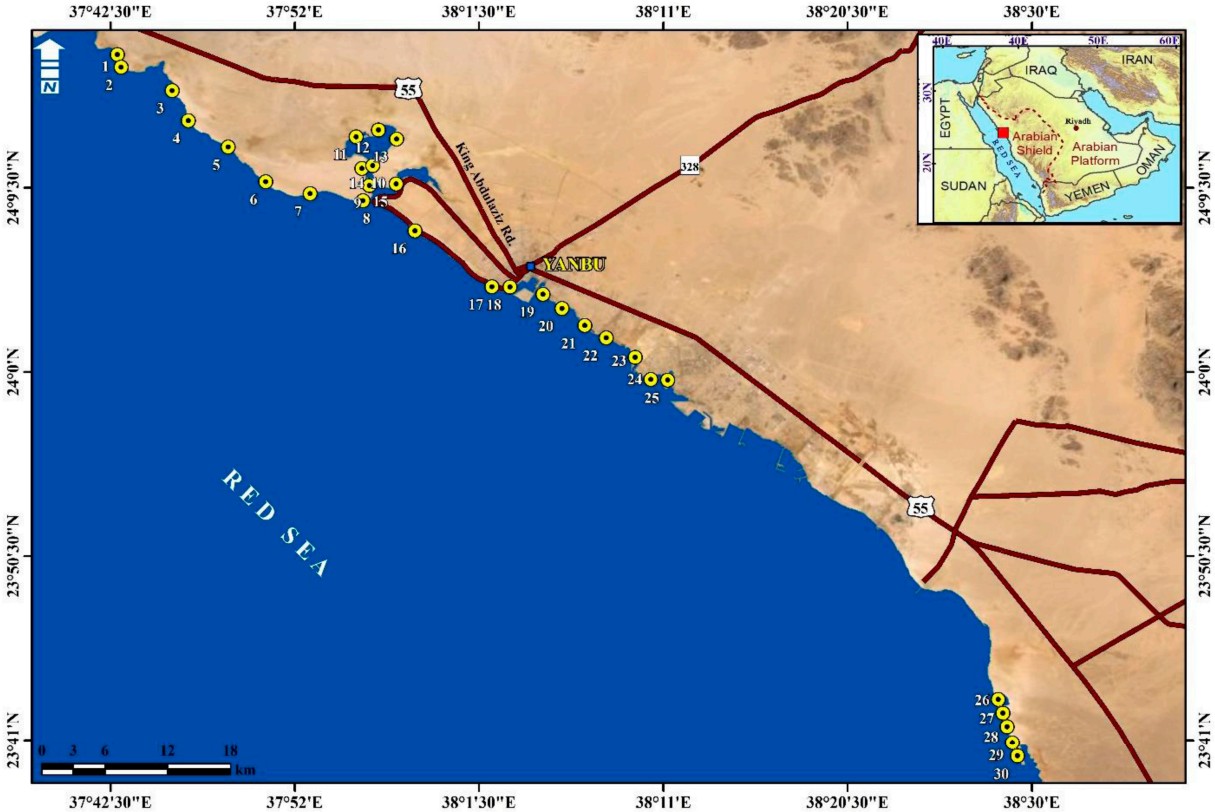

**Figure 1.** Study area map showing seawater sampling locations.

*2.2. Pollution Indices and Multivariate Analyses*

Pollution assessment was achieved by calculating the contamination index (Cd) and the heavy metal pollution index (HPI) [6]. The contamination index, which sums the combined effects of several domestic water quality parameters [6,38], is calculated as follows:

$$C_d = \Sigma CF \tag{1}$$

$$CF = C\ (heavy\ mental)/C\ (Background)$$

where CF is defined as the contamination factor, *C* (*heavy metal*) is the analytical value, and *C* (*Background*) is the upper permissible concentration limit for the ith component. Three categories of $C_d$ are recognized [38]: low contamination ($C_d < 4$), medium contamination ($C_d = 4$–8), and high contamination ($C_d > 8$).

HPI is determined using the expression below [38]:

$$HPI = \Sigma QiWi/\Sigma Wi \tag{2}$$

where Qi is the sub-index and Wi is the unit weight of the ith parameter.

$$Qi = \Sigma([Mi\ (-)\ Ii]/(Si - Ii)) \tag{3}$$

where Mi, Ii, and Si represent the targeted heavy metal, ideal, and standard ith parameter values, respectively. Standard WHO values provided the standard permissible (Si) and the highest desirable(Ii) values for each parameter. HPI is categorized into three classes [39,40]: low pollution (HPI < 5), medium pollution (HPI = 5–10), and high pollution (HPI > 10). To determine the potential sources of HMs in the analyzed seawaters, R and Q mode dendrogram, principal component analysis, and Pearson's correlation coefficient were carried out using Windows SPSS 16.0.

## 3. Results and Discussion

### 3.1. Distribution and Contamination of Heavy Metals

Supplementary Table S1 shows coordinates for sampling locations, heavy metals concentrations, and calculated pollution indices in the study area. Minimum, maximum, and average HMs values, as well as maximum admissible limits and standard deviation, were presented in Table 1. Ranges of HMs values can be arranged in the following order: Ni (1.30–6.80 µg/L) > As (2.50–8.10 µg/L) > Cu (1.10–3.80 µg/L) > Zn (0.30–3.20 µg/L) > Al (0.70–1.60 µg/L) > Fe (0.40–1.80 µg/L) > Cr (0.10–1.20 µg/L) > Mn (0.21–0.42 µg/L) > Cd (0.22–0.45 µg/L) > Pb (0.09–0.43 µg/L) > Sb (0.12–0.38 µg/L) > Co (0.07–0.26 µg/L) > Hg (0.02–0.10 µg/L). Deciphering HMs origins in seawater is a complex task due to the fact that these HMs are not persistent in seawater but are rather mobilized and influenced by water dynamics [6,41]. A fluctuating pattern for the distribution of measured HMs was noted (Figures 2 and 3) with some samples containing higher levels of different HMs, for example, sample 1 (Zn and Al), sample 2 (Sb), sample 7 (Pb), sample 8 (Hg and Mn), sample 10 (Cr), sample 14 (As), sample 15 (Fe), sample 20 (Cd), sample 23 (Cu), and sample 24 (Co and Ni). This pattern indicates anthropogenic HMs origin, probably from nearby agricultural and industrial waste [42].

**Table 1.** Minimum, maximum, and average HMs values (µg/L) along with standard deviation and maximum admissible concentration (MAC) [43,44].

|     | Minimum | Maximum | Average | Std. Deviation | MAC |
|-----|---------|---------|---------|----------------|-----|
| As  | 2.50    | 8.10    | 4.297   | 1.426          | 10  |
| Cd  | 0.22    | 0.45    | 0.309   | 0.061          | 3   |
| Co  | 0.07    | 0.26    | 0.144   | 0.058          | 5   |
| Cr  | 0.10    | 1.20    | 0.723   | 0.281          | 50  |
| Cu  | 1.10    | 3.80    | 2.447   | 0.765          | 2000|
| Fe  | 0.40    | 1.80    | 0.983   | 0.361          | 200 |
| Hg  | 0.02    | 0.10    | 0.058   | 0.023          | 6   |
| Mn  | 0.21    | 0.42    | 0.328   | 0.058          | 50  |
| Pb  | 0.09    | 0.43    | 0.276   | 0.094          | 10  |
| Ni  | 1.30    | 6.80    | 4.424   | 1.690          | 20  |
| Zn  | 0.30    | 3.20    | 1.667   | 0.901          | 40  |
| Sb  | 0.12    | 0.38    | 0.238   | 0.075          | 20  |
| Al  | 0.70    | 1.60    | 1.133   | 0.206          | 200 |

It is noted in this study that Cd and Ni average values far exceed those reported from coastal areas worldwide, except the Caspian coast, Iran [45], and North Atlantic and North Pacific [46], respectively (Table 2). Moreover, average As is also higher than those reported from many seawaters, except North Atlantic and North Pacific [46], and Tarut Island, Arabian Gulf [14]. The average values of Zn, Ni, Co, Cr, Cu, and Hg fluctuate in comparison with surface seawaters included in Table 2. Moreover, average HMs values measured in sediments from the Yanbu area [35] were higher than those recorded in seawater samples, except Cd. HMs obtained from Yanbu sediments ranged from 7520 times in aluminum to 0.97 times in cadmium (Table 3). However, sediment always acts as functional sinks for HMs driven from different sources [47], as well as the change in environmental conditions, such as pH, redox potential, and salinity, sediments can release more heavy metals into the seawater [9,48].

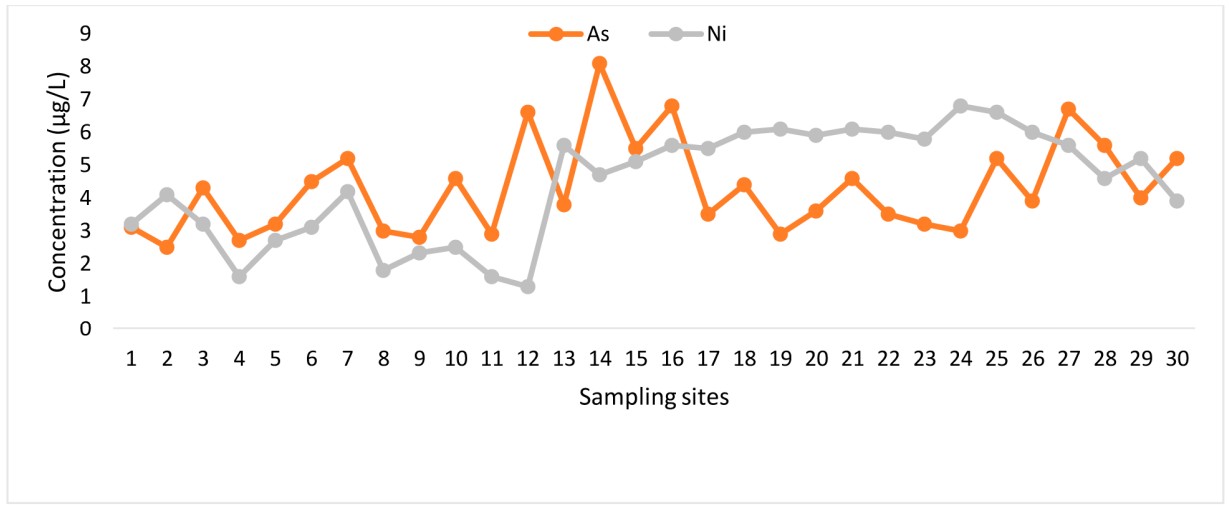

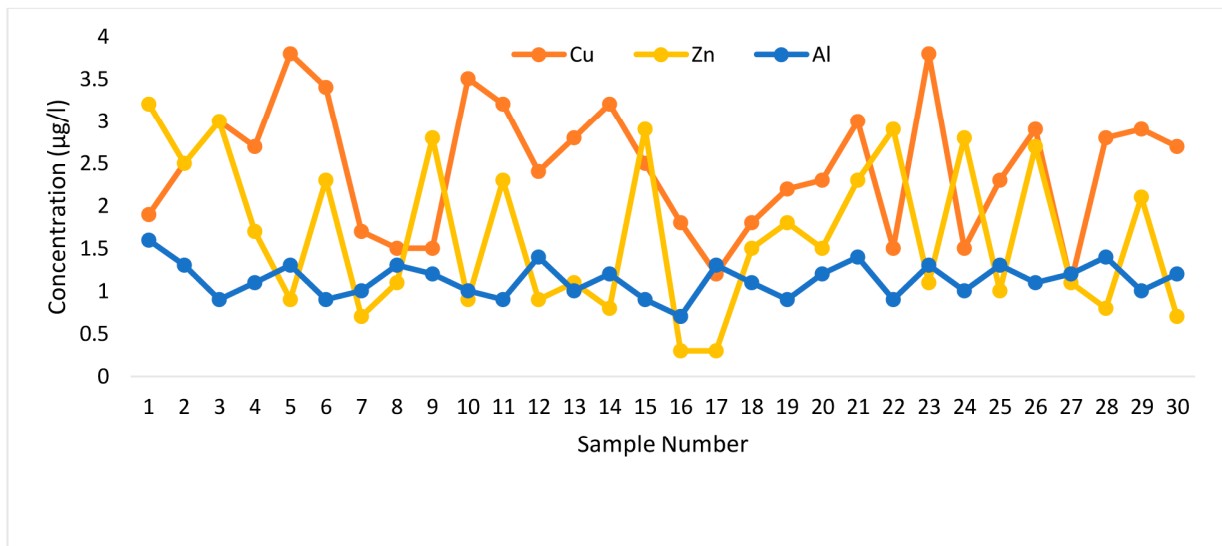

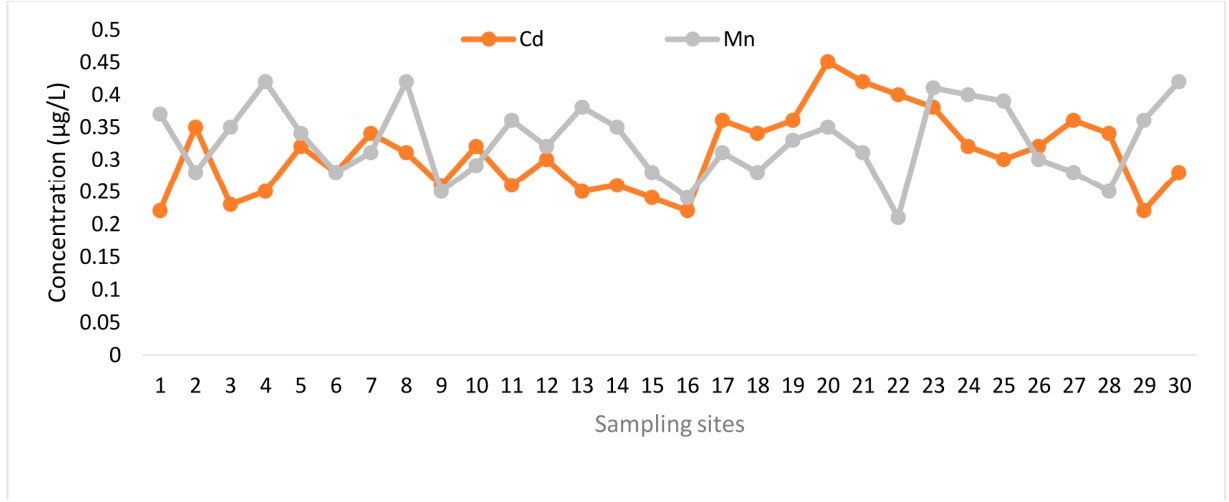

**Figure 2.** Distribution of As, Cu, Ni, Zn, Al, Cd, and Mn of the analyzed seawaters.

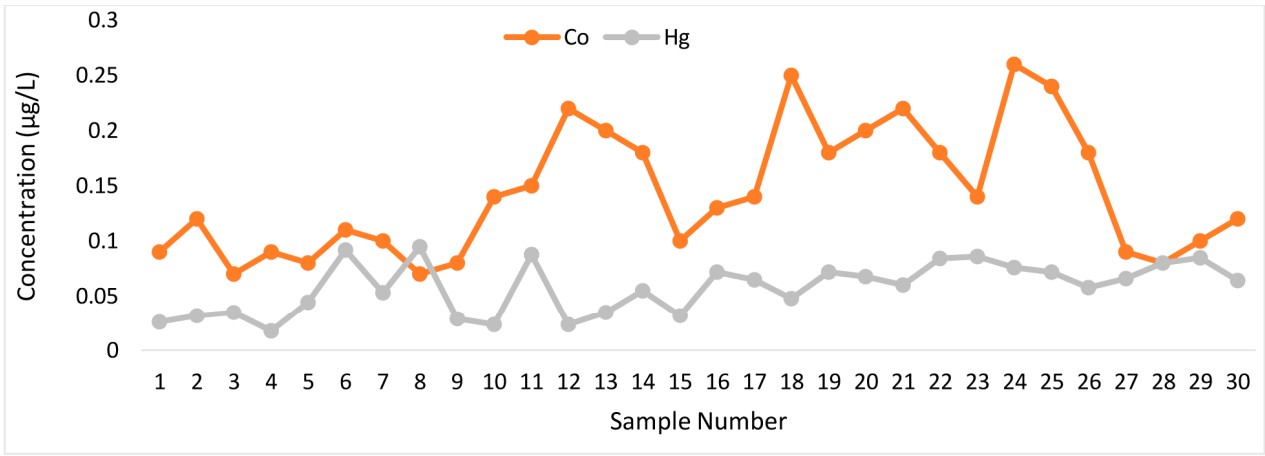

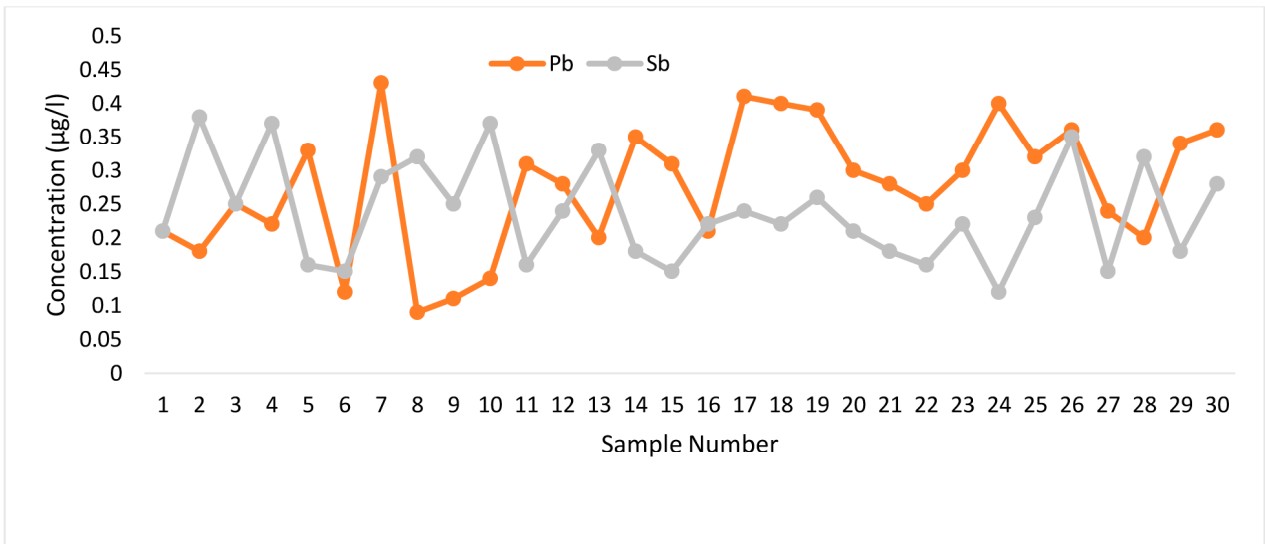

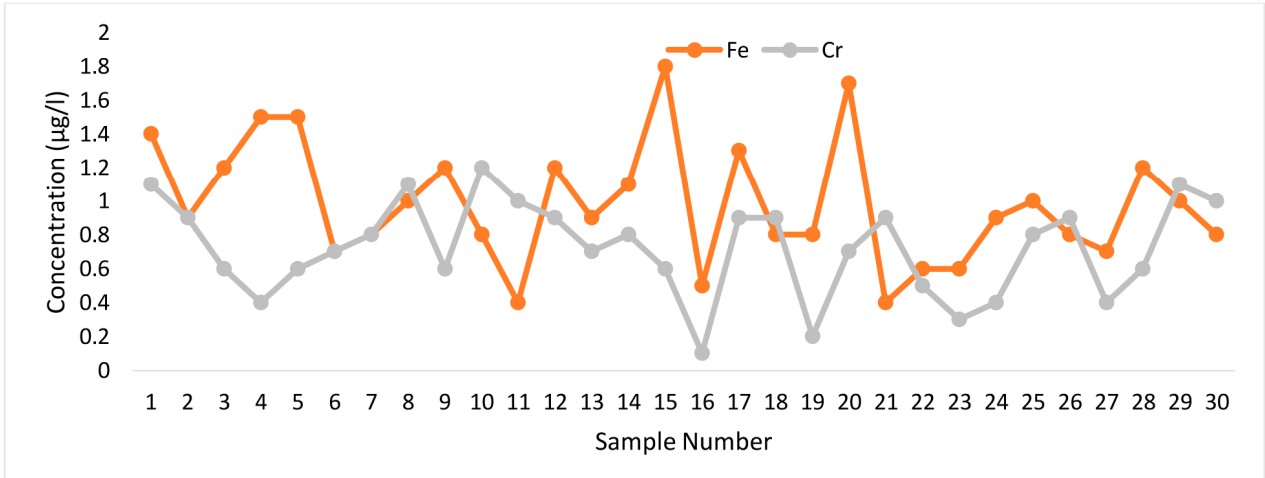

**Figure 3.** Distribution of Co, Hg, Pb, Sb, Fe, and Cr in seawater samples.

**Table 2.** HMs levels (μg/L) reported in this study as compared to values reported worldwide.

| | Pb | Cd | Zn | Ni | Co | Cr | Cu | Hg | As | Reference |
|---|---|---|---|---|---|---|---|---|---|---|
| Yanbu coastline, Red Sea, Saudi Arabia | 0.28 | 0.31 | 1.67 | 4.42 | 0.14 | 0.72 | 2.44 | 0.058 | 4.30 | Present study |
| Red Sea-Gulf of Aqaba, Saudi Arabia | 1.31 | 0.05 | 5.51 | 2.45 | 0.31 | 0.26 | 2.34 | 0.008 | 2.43 | [6] |
| Rosetta coast, Mediterranean Sea, Egypt | 0.426 | - | 1.694 | 1.92 | - | 0.133 | - | - | 0.30 | [8] |
| Al-Khobar, Arabian Gulf, Saudi Arabia | 0.04 | 0.11 | 16.21 | 4.36 | 0.36 | 1.38 | 5.24 | 0.68 | 2.41 | [9] |
| Tarut Island, Saudi Arabia | 0.48 | 0.03 | 0.97 | - | 2.06 | 12.95 | 2.65 | 0.30 | 11.13 | [14] |
| Caspian coast, Iran | 1.67 | 0.27 | 16.94 | 9.93 | 1.65 | - | 5.02 | - | - | [45] |
| North Atlantic | 125 | 5.5 | 0.15 | 2 | 159 | 3.5 | 1.15 | 1–7 | 20 | [46] |
| North Pacific | 32 | 5.5 | 0.15 | 2 | 27 | 3 | 0.9 | 0.5–10 | 20 | [46] |
| Al-Khafji, Arabian Gulf, Saudi Arabia | 0.28 | 0.07 | 1.53 | 4.40 | 0.23 | 2.44 | 2.44 | 0.06 | 1.74 | [47] |
| Average oceanic concentration | 0.001 | 0.07 | 0.4 | - | - | 0.33 | 0.12 | - | - | [49] |
| Gulf of Aqaba | 0.32 | 0.57 | 0.24 | 0.22 | 0.17 | - | 0.14 | - | - | [50] |
| Red sea coast, Egypt | 0.03 | 0.06 | 5.5 | 0.76 | 0.03 | 0.18 | 0.97 | - | - | [51] |
| Gulf of Aqaba, Saudi Arabia | 0.20 | 0.03 | 3.32 | - | 0.24 | 0.96 | 6.18 | 0.06 | 0.82 | [51] |

**Table 3.** Comparison between HMs concentration in the seawater (μg/L) and sediment (μg/g) of the study area.

| Element | Seawater Averages | Sediment Averages | Sediment/Seawater |
|---|---|---|---|
| Al | 1.14 | 8573 | 7520 times |
| Fe | 0.99 | 5895 | 5955 times |
| Mn | 0.33 | 192 | 581.82 times |
| Zn | 1.67 | 80.40 | 48.14 times |
| Co | 0.15 | 8.29 | 55.27 times |
| Cr | 0.72 | 27.11 | 37.65 times |
| Pb | 0.28 | 7.72 | 27.57 times |
| Cu | 2.45 | 35.87 | 14.64 times |
| Hg | 0.06 | 0.33 | 5.50 times |
| Ni | 4.43 | 23.50 | 5.30 times |
| Sb | 0.24 | 0.50 | 2.08 times |
| As | 4.30 | 6.83 | 1.59 times |
| Cd | 0.31 | 0.30 | 0.97 times |

The degree of contamination ($C_d$) and the heavy metal pollution index (HPI) are two quantitative methods commonly used in the assessment of seawater quality [6]. All calculated HMs average concentrations were less than the maximum admissible concentration values published by the World Health Organization [52]. The spatial distribution of pollution indices (HPI and $C_d$), although clearly fluctuating, corresponds very well with HMs spatial distribution in the Yanbu area (Figure 4). Based on the classification of pollution indices, HPI levels ranged from 4.51 in sample 4 to 9.59 in sample 27, with 7.45 being the average (Supplementary Table S1). All seawater samples are classified as medium pollution, except samples 4 and 5 (6.66%), which fell under low pollution. Samples with higher HM values were also found to be characterized by higher HPI levels. Cadmium levels ranged from 1.04 in sample 9 to 2.24 in sample 14 in a bay (Figure 4), with an average of 1.67. Results of Cd indicated low contamination for all seawaters investigated.

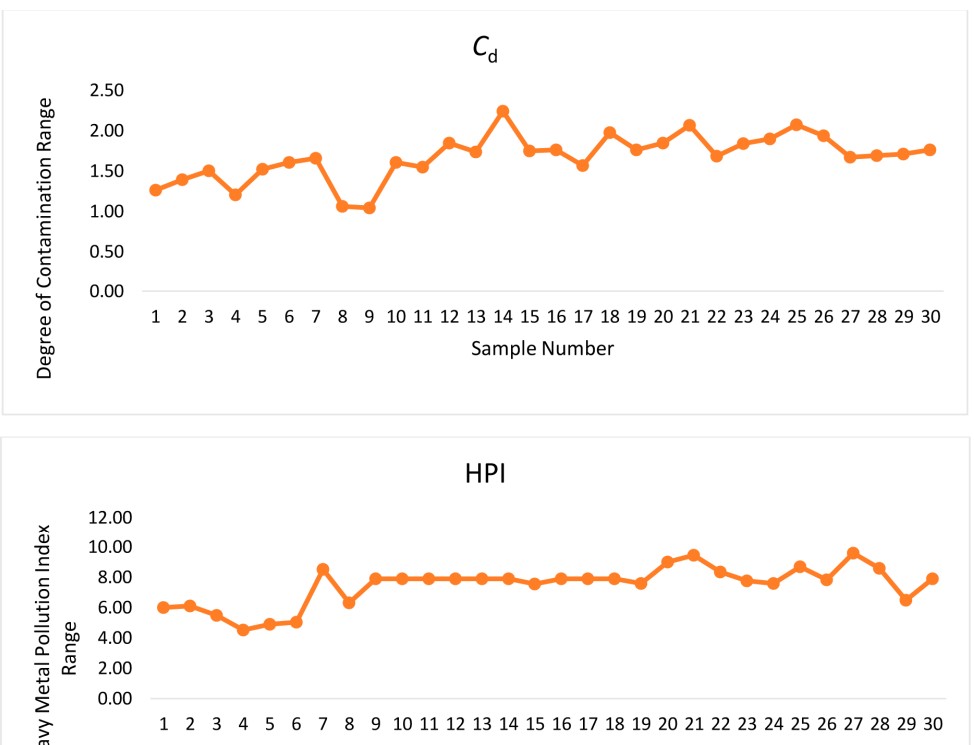

**Figure 4.** Distribution of degree of contamination ($C_d$) and heavy metal pollution index (HPI) of Yanbu seawater samples.

### 3.2. Sources Analysis of HMs

Hierarchical cluster analysis (HCA), correlation analysis (CA), and principal component analysis (PCA) are the most common multivariate statistical analyses used to determine HMs sources and for the interpretation of datasets [53,54]. Cluster analysis is a technique used in classifying HMs based on their similarity or dissimilarity [55]. The 13 HMs are categorized into 2 clusters based on R mode HCA (Figure 5). The first cluster includes Cd, Mn, Pb, Sb, Co, Hg, Fe, Al, Cr, Cu, and Zn. Al, Fe, and Mn are among the most preferred geogenic indicators [17] and their observed close association with Cd, Pb, Sb, Co, Hg, Cr, Cu, and Zn, which are of mainly anthropogenic sources [56,57], indicated mixed geogenic and anthropogenic sources for the elements of cluster 1. The second cluster includes As and Ni and, therefore, might indicate anthropogenic sources, possibly derived from Yanbu industrial city. However, it should be noted that As in the environment may also be derived from natural sources such as atmospheric deposition and volcanic activity, in addition to anthropogenic activities such as mining, domestic wastewater, refining processes, and agricultural production [42,58–60]. In the human body, As toxicity is mainly associated with skin hyperpigmentation and hyperkeratosis, and metabolic disorders [61].

Factors and sources of HMs in environmental studies are known by Pearson's correlation [6,62], which is used to detect similar sources of heavy metals with positive correlation; that is, to evaluate the strength of the linear association between two variables [6,57,62,63]. A positive correlation between Co-Ni and a low positive correlation between Ni-Cd, Ni-Pb, Ni-Hg, Al-Cr, Co-Cd, and Mn-Cu was shown by Pearson correlation coefficient (Table 4), suggesting that their concentrations were controlled by the same geochemical processes [28]. The correlation of the well-known geogenic elements Al and Mn with Cr and Cu suggests that mainly natural sources influenced their concentrations in the coastal water. Their association with oxides and hydroxides of Mn and Al implicated rock weathering as a natural contributing source, besides the anthropogenic activities [15,64,65]. The positive correlation between Ni-Co, and Ni-Hg, suggests anthropogenic sources, possibly from agricultural

and industrial activities around the study area [29,66] apart from their natural sources. Moreover, the positive correlation between Zn-Cd, Zn-Cr, and Zn-Pb suggests that these metals were anthropogenically sourced, probably from agricultural-related activities [30,45]. Unmanaged landfilling and domestic wastewaters near the seawater are possible sources of the elevated concentrations of Cd and Pb [67,68]. Hg and Cd transfer from marine sea foods to the human body, causing destruction in the central nervous system and metabolic disorders, respectively [61]. On the other hand, HMs pairs, such as Sb-As, Sb-Cd, Sb-Co, Sb-Cu, Sb-Hg, Sb-Pb, Sb-Ni, and Sb-Zn, display negative correlations among them (Table 4). The same correlation pattern was also noted between pairs of Ni-Cd, Ni-Co, Ni-Cu, Ni-Hg, Ni-Mn, and Ni-Zn. The implication of such negative correlations is that there is more than one source of contamination in the area [32,47].

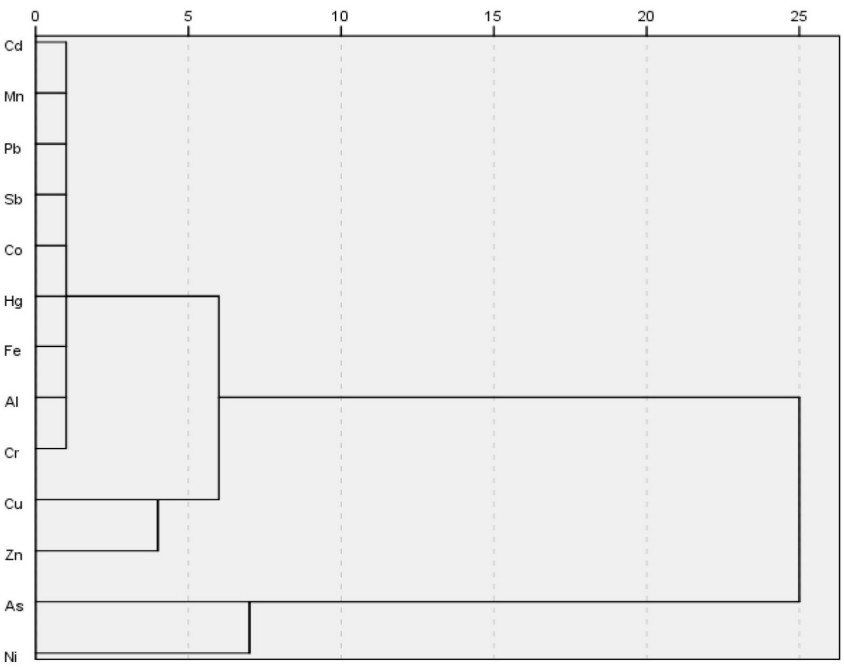

**Figure 5.** R mode HCA of the investigated HMs.

**Table 4.** Correlation matrices of the investigated HMs.

|  | As | Cd | Co | Cr | Cu | Fe | Hg | Mn | Pb | Ni | Zn | Sb | Al |
|---|---|---|---|---|---|---|---|---|---|---|---|---|---|
| As | 1 | | | | | | | | | | | | |
| Cd | −0.154 | 1 | | | | | | | | | | | |
| Co | 0.092 | **0.368 *** | 1 | | | | | | | | | | |
| Cr | −0.057 | −0.096 | −0.001 | 1 | | | | | | | | | |
| Cu | 0.006 | −0.134 | −0.072 | 0.137 | 1 | | | | | | | | |
| Fe | −0.076 | −0.174 | −0.271 | 0.041 | −0.033 | 1 | | | | | | | |
| Hg | −0.015 | 0.288 | 0.097 | −0.134 | −0.059 | −0.475 ** | 1 | | | | | | |
| Mn | −0.261 | −0.188 | 0.067 | 0.174 | 0.234 | 0.153 | 0.057 | 1 | | | | | |
| Pb | 0.112 | 0.212 | **0.443 *** | −0.056 | −0.064 | 0.000 | 0.132 | 0.186 | 1 | | | | |
| Ni | 0.132 | **0.413 *** | **0.536 **** | −0.340 | −0.231 | −0.219 | **0.357** | −0.143 | **0.495 **** | 1 | | | |
| Zn | −0.462 * | −0.142 | −0.027 | 0.029 | −0.019 | 0.066 | −0.132 | −0.119 | −0.146 | −0.007 | 1 | | |
| Sb | −0.195 | −0.025 | −0.190 | 0.228 | 0.086 | 0.056 | −0.385 * | 0.094 | −0.295 | −0.267 | −0.222 | 1 | |
| Al | −0.052 | 0.266 | −0.022 | **0.380 *** | −0.012 | 0.291 | −0.183 | 0.242 | −0.075 | −0.136 | −0.133 | 0.140 | 1 |

* Correlation is significant at the 0.05 level (2-tailed). ** Correlation is significant at the 0.01 level (2-tailed). Bold numbers mean positive correlation.

Q mode HCA categorized the studied 30 localities into 2 clusters based on their similarity and HMs characteristics (Figure 6). The first cluster is composed of samples 1–6 and 8–12, having the highest values of Zn, Sb, Al, Hg, Mn, and Cr (samples 1, 2, 8, and 10). The second cluster includes samples 7 and 13–30, which have the highest amounts of Pb, As, Fe, Cd, Cu, Ni, and Co (samples 7, 14, 15, 20, 23, 24). Applying principal component analysis

(PCA) on the HMs, which is an analytical tool based on the relationship between different variables [69], five principal components are generated. These principal components account for 22.36%, 13.35%, 11.30%, 10.20, and 9.89%, respectively, and cumulatively explain 67.10% of the total variance (Table 5). The first component shows high positive loading for Co, Hg, Pb, and Ni, while the third shows positive loading for Zn, which reflects anthropogenic sources from industrial activities and farm drainage in the study area through atmospheric input [55,69–72]. Moreover, traffic emissions are also a common source for Pb and Zn [72]. Agricultural practices (e.g., uses of pesticides, fungicides, and phosphate fertilizers) are implicated in arsenic and Cu contamination [65,73]. The second component displays positive loading for Mn and Al, which are mainly derived from geogenic sources, reflected water–sediment interaction [4,74]. The fourth component accounts for positive loading for Cu and Mn. Positive loading for Fe and Ni is noted in the fifth component, which may indicate that these metals were sourced from natural (i.e., rock weathering) and anthropogenic and industrial sources [69,75]. In spite of Cu being an essential trace element in the body, excessive copper can lead to Cu- metabolic disorders and liver cancer mortality.

The findings of our study revealed that some seawaters (samples 1, 2, 7, 8, 10, 14, 15, 20, 23, and 24) had higher HMs, as shown by a strong agreement between hierarchical cluster analysis and contamination indices. Therefore, the integration of multivariate analysis contamination indices in seawater quality assessment regarding HMs is a beneficial and adaptable approach [3]. Furthermore, this study indicated that the uncontrolled discharge of domestic, industrial, and agricultural wastewater and chemicals into the seawaters along the Red Sea coast result in increased metal levels, leading to the deterioration of the marine environment and health risks for human beings. Accordingly, continuous monitoring of HM levels in marine seawaters encompasses physical, chemical, and biological aspects is needed [76].

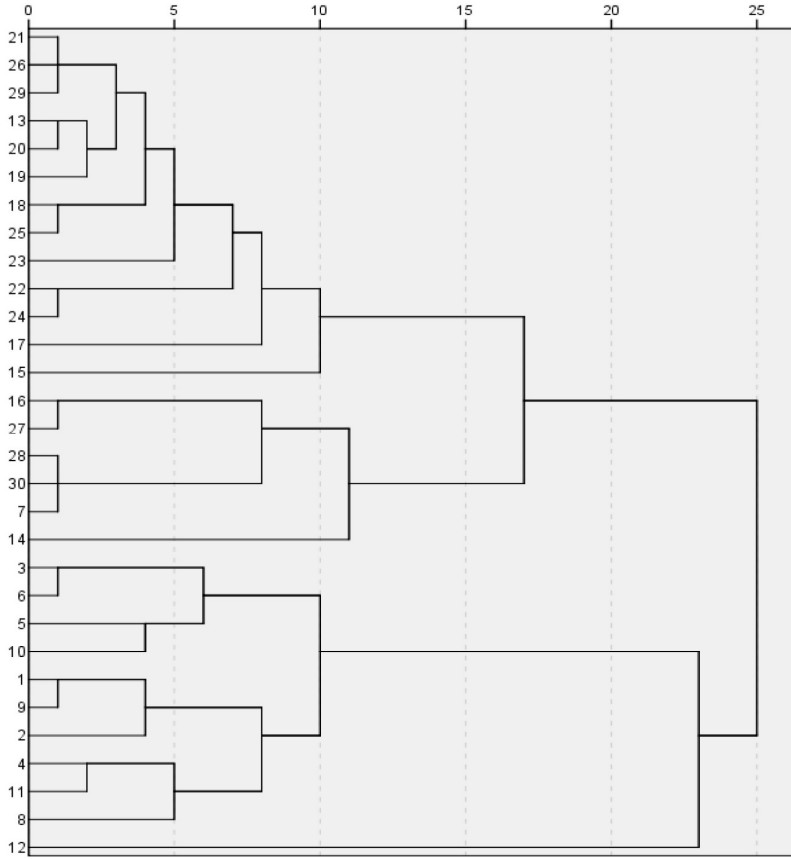

**Figure 6.** Q mode HCA of Yanbu seawater samples.

**Table 5.** Principal component loadings and variance percentage for the five components.

|  | Component | | | | |
|---|---|---|---|---|---|
|  | **1** | **2** | **3** | **4** | **5** |
| As | 0.221 | −0.095 | −0.848 | −0.118 | 0.202 |
| Cd | **0.512** | 0.410 | 0.193 | −0.286 | −0.480 |
| Co | **0.629** | 0.393 | 0.097 | 0.006 | 0.106 |
| Cr | −0.406 | 0.470 | 0.015 | 0.164 | −0.146 |
| Cu | −0.263 | 0.102 | −0.119 | **0.620** | 0.069 |
| Fe | −0.469 | 0.186 | 0.084 | −0.449 | **0.542** |
| Hg | **0.587** | −0.075 | 0.052 | 0.458 | −0.314 |
| Mn | −0.209 | **0.540** | 0.145 | **0.536** | 0.307 |
| Pb | **0.570** | 0.390 | −0.006 | 0.050 | **0.502** |
| Ni | **0.826** | 0.102 | 0.095 | −0.163 | 0.087 |
| Zn | −0.116 | −0.353 | **0.793** | −0.034 | 0.166 |
| Sb | −0.506 | 0.238 | −0.133 | −0.105 | −0.430 |
| Al | −0.286 | **0.705** | 0.030 | −0.281 | −0.139 |
| % of Variance | 22.36 | 13.35 | 11.30 | 10.20 | 9.89 |
| Cumulative % | 22.36 | 35.71 | 47.01 | 57.21 | 67.10 |

Bold numbers mean positive correlation.

## 4. Conclusions

This study used pollution indices (Cd and HPI) to assess the quality of seawaters along the Yanbu coastal area for heavy metals pollution and to highlight the potential pollution sources via the utilization of multivariate statistical analyses. The following findings were obtained:

1.  The analyzed 30 seawater samples display a descending order for the contained HMs averages (μg/L): Ni > As > Cu > Zn > Al > Fe > Cr > Mn > Cd > Pb > Sb > Co > Hg.
2.  The average values of Ni, Cd, and As were higher than those reported from many worldwide coastal areas.
3.  The distribution of pollution indices was mostly compatible with the HMs spatial distribution. Based on HPI, 93.34% of the analyzed samples were within medium pollution, and 6.66% of them were under low pollution. Levels of Cd indicated low contamination for all seawater samples.
4.  Multivariate statistical analyses indicated that Cd, Co, Hg, Zn, and Ni, might originate from anthropogenic industrial sources linked to brass and bronze, paints, galvanizing, paint pigments, batteries, cable covering, rubber, and sacrificial anodes on marine water crafts.
5.  On the other hand, Mn, Pb, Sb, Fe, Al, Cr, and Cu, might be originated from combined lithogenic and human sources.

**Supplementary Materials:** The following supporting information can be downloaded at: https://www.mdpi.com/article/10.3390/w15010201/s1, Table S1: The coordinates of the sampling sites, pH, electrical conductivity (EC, mS/cm), salinity (Sal, ppt) and the HM analyses (μg/L), and the pollution indices of the 30 surface seawater samples from the area.

**Author Contributions:** Conceptualization, A.S.E.-S. and M.Y.; methodology, A.S.E.-S., M.Y. and M.H.A.-H.; data curation, A.S.E.-S. and M.Y.; writing—original draft preparation, A.S.E.-S. and M.Y.; writing—review and editing, A.S.E.-S., M.Y. and M.H.A.-H. All authors have read and agreed to the published version of the manuscript.

**Funding:** This research was funded by the Deputyship for Research & Innovation, Ministry of Education in Saudi Arabia for funding this research work through the project no. (IFKSURG-2-446).

**Data Availability Statement:** Not applicable.

**Acknowledgments:** The authors extend their appreciation to the Deputyship for Research & Innovation, Ministry of Education in Saudi Arabia for funding this research work through the project no. (IFKSURG-2-446). In addition, the authors would like to thank the anonymous reviewers for their valuable suggestions and constructive comments.

**Conflicts of Interest:** The authors declare no conflict of interest.

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
