# Peer review of "Water Quality Assessment and Environmental Impact of Heavy Metals in the Red Sea Coastal Seawater of Yanbu, Saudi Arabia"

_water, doi:10.3390/w15010201_

Round 1

Reviewer 1 Report

dear editor

i have read this manuscript. its theme is interesting, how ever i have proposed some points.

best regards

Author Response

Comment

Response

In abstract please clarify what are the classes of water samples based on the WHO?

It has been mentioned that the HPI is categorized into three classes [4, 37]: low pollution (HPI < 5), medium pollution (HPI = 5–10), and high pollution (HPI > 10). (Lines 106-108)

Accordingly, the classes of water samples have been outlined in the results section: “All seawater samples are classified as medium pollution, except samples 4 and 5 (6.66 %) which fell under low pollution.” (Lines 188, 189)

The classes of water samples have also been mentioned in the abstract: “heavy metal pollution (HPI) revealed medium contamination levels in 28 samples and low levels in two samples.” (Lines 20-21)

In introduction section, you need to discuss about the importance of filed and numerical studies of water quality rivers and lakes and so on.

Relevant references with regard to water quality in rivers and lakes have been added.

In results and discussion other diagram of water quality should be prepared to this end you can refer to the following articles:

There are not attached articles

Reviewer 2 Report

Dear authors, thank you for submitting the manuscript to the journal of Water. Its topic is very interesting. However, the current version of the paper suffers from a number of weaknesses related to the empirical strategy used. I have the following comments/questions for the authors:

Abstract

·         Add important results in the abstract section.

·         The authors ought to re-write the abstract so that it briefly presents the problem at hand, objectives of the study, methods used to achieve the objectives in logical order. Also, abstract section should be completed with the results of the study.

Introduction

·         In introduction chapter please focus on problem generally, on the basis of examples in the whole World, not your study area.

·         Add some facts and figures of surface water quality around the globe in your introduction.

·         Add some recent article to make your introduction more attractive and strong. I propose to add this survey method in the overview section of the introduction section, based on the latest literature. Please replace old citations (if it is possible) or add citations of newest literature.

https://doi.org/10.3390/w13162258

https://doi.org/10.3390/w14071131

Methods

Location and sampling

·         Please give detailed information on water samplers (e.g., accuracy, manufacturer).

·         Sampling locations were selected carefully along the Red Sea coast of Saudi Arabia to have a good representation of the spatial variability of quality indicators across-section of water quality monitoring. What criteria where analyzed to select this locations?

·         Please provide detailed detection methods and quality control results?

·         Please support your methods by providing appropriate references or give the guidelines used to analyze the water quality parameters.

·         Please support your methods by providing appropriate references or give the guidelines used and equations.

·         How did you do quality control (QC) and quality assurance (QA) on the obtained data to validate the conclusions?

Results and discussion

·         You should think how transformational the research is likely to be should be made so that the outcome of the work will have an impact on the community/society facing given sustainability related challenges?

·         Write the practical applications of your work in a separate section, before the conclusions and provide your good perspectives.

·         What are the likely research impacts of this work globally, nationally and locally?

·         What long-term impacts will it have on environmental protection and the wider public or the field following the completion of the research?

·         I suggest representing the data results of principal component analysis (PCA) in figure explaining the effect of metals on the water quality.

Conclusion

·         Concise the text in conclusion and add future work in order to recommend your work. Shorten the length of each and every paragraph by adding only relevant and major findings in your study. 

Please respond to all of those comments in the revised manuscript by pointing out precisely and concisely on which page and in which line you have incorporated your response one by one.

Author Response

Comment

Response

Abstract

•            Add important results in the abstract section.

•            The authors ought to re-write the abstract so that it briefly presents the problem at hand, objectives of the study, methods used to achieve the objectives in logical order. Also, abstract section should be completed with the results of the study.

The importance of the study and the problem have been highlighted in the abstract (Lines 10-12). The objectives of the study are listed in the abstract (Lines 12-14). Methods have been listed (Lines 15-16).  All the results of the study have been mentioned in the abstract (17-25).

Introduction

•            In introduction chapter please focus on problem generally, on the basis of examples in the whole World, not your study area.

•            Add some facts and figures of surface water quality around the globe in your introduction.

•            Add some recent article to make your introduction more attractive and strong. I propose to add this survey method in the overview section of the introduction section, based on the latest literature. Please replace old citations (if it is possible) or add citations of newest literature.

https://doi.org/10.3390/w13162258

https://doi.org/10.3390/w14071131

As suggested by the reviewer, recent references (Gad et al., 2021, 2022) have been added to the introduction. (Line 31)

Location and sampling

•            Please give detailed information on water samplers (e.g., accuracy, manufacturer).

•            Sampling locations were selected carefully along the Red Sea coast of Saudi Arabia to have a good representation of the spatial variability of quality indicators across-section of water quality monitoring. What criteria where analyzed to select this locations?

•            Please provide detailed detection methods and quality control results?

•            Please support your methods by providing appropriate references or give the guidelines used to analyze the water quality parameters.

•            Please support your methods by providing appropriate references or give the guidelines used and equations.

•            How did you do quality control (QC) and quality assurance (QA) on the obtained data to validate the conclusions?

•            Sampling locations were selected carefully along the Red Sea coast of Saudi Arabia either nearby the locations of the human activities; agricultural, fishing, industrial, and urbanization activities.

•            The calibration curves of the 6 elements: Cr, Cd, Cu, Pb, Co, and Ni were obtained using the blank and three working standards 0, 50, 100, and 200 μg/L (Panreac, 766333. 1208). For As and Zn elements, using the blank and three working standards 0, 50, 100, and 200 μg/L (Aristar grade, BDH laboratory supplies, England for the heavy metals). Calibration curves showed an excellent linearity for all elements.

Results and discussion

•            You should think how transformational the research is likely to be should be made so that the outcome of the work will have an impact on the community/society facing given sustainability related challenges?

•            Write the practical applications of your work in a separate section, before the conclusions and provide your good perspectives.

•            What are the likely research impacts of this work globally, nationally and locally?

•            What long-term impacts will it have on environmental protection and the wider public or the field following the completion of the research?

•            I suggest representing the data results of principal component analysis (PCA) in figure explaining the effect of metals on the water quality.

A separate section has been written at the end of results and discussion section indicated the practical applications of our work. As well as the outcome of this work on the community.

Conclusion

•            Concise the text in conclusion and add future work in order to recommend your work. Shorten the length of each and every paragraph by adding only relevant and major findings in your study.

This study used pollution indices (Cd and HPI) to assess the quality of seawaters along the Yanbu coastal area for heavy metals pollution and to highlight the potential pollution sources via the utilization of multivariate statistical analyses. The following findings were obtained:

1.       The analyzed thirty seawater samples display a descending order for the contained HMs averages (μg/L): Ni > As > Cu > Zn > Al > Fe > Cr > Mn > Cd > Pb > Sb > Co > Hg.

2.        The average values of Ni, Cd, and As were higher than those reported from many worldwide coastal areas.

2. The distribution of pollution indices was mostly compatible with the HMs spatial distribution. Based on HPI, 93.34% of the analyzed samples were within medium pollution and 6.66 % of them were under low pollution. Levels of Cd indicated low contamination for all seawater samples.

3. Multivariate statistical analyses indicated that Cd, Co, Hg, Zn, and Ni, might originated from anthropogenic industrial sources linked to brass and bronze, paints, galva-nizing, paint pigments, batteries, cable covering, rubber and sacrificial anodes on marine water crafts.

4. On the other hand Mn, Pb, Sb, Fe, Al, Cr, and Cu, might be originated from combined lithogenic and human sources.

Reviewer 3 Report

The authors performed the water quality assessment and environmental impact evaluation of heavy metals in the Red Sea coastal seawater of Yanbu, Saudi Arabia. Overall, the study is methodological sound, with promising results and discussion. However, prior to further consideration, some comments need to be addressed.

(i) Title: Environmental Impact evaluation seems to be slightly confused with the term environmental impact assessment of heavy metal. However, in this study, the impacts on the environment seem not much discussed. The authors seem just applied HPI and other indexes for heavy metal evaluation. Therefore, the title requires revison.

(ii) Introduction: Huge modifications are necessary for the introduction section. The research gaps and significance of the study are not shown. Moreover, the authors should provide more precise research objectives in this study. Also, please highlight some incidents of heavy metal pollution that occurred in the region which drove you to perform the study.

(iii) The reasons spatial distribution of the heavy metals were not clearly illustrated, for instance why a certain area has higher pollution than other places?

(iv) Discuss the upstream and downstream differences in this study. I would suggest to incorporate the landuse type into the discussion. 

(v).  So what are the proposed mitigations? What should the government do for proper water quality management? Effects from changing land use or anthropogenic activities? The difference between upstream or downstream is not provided.

(vi).  Some relevant literature is missing in this study.

a.      doi:10.1007/s10661-020-08543-4.

b.      doi: 10.1007/s10661-021-09202-y

(vii) I would suggest replacing Table 2 with a box plot for better illustration. Also, for the MAC, it is following the WHO values? Do you have your own standards/limits in Saudi Arabia?

(viii) Results in section 3.2 seems to be a very general discussion, and not region specific. I cannot follow the main reasons for the causes. Please revise.

(ix) Conclusion section seems to be a repetition of the results section. Huge modifications are required. Please provide insights into this study and what can be further done in the future.

(x) I do not find the significance of determining the correlation between parameters in this study. Please provide more details. I understand that you would like to include different statistical analyses for enriching the study, however, the flow is not there. Please revise.

Author Response

Comment

Response

(i) Title: Environmental Impact evaluation seems to be slightly confused with the term environmental impact assessment of heavy metal. However, in this study, the impacts on the environment seem not much discussed. The authors seem just applied HPI and other indexes for heavy metal evaluation. Therefore, the title requires revison.

As suggested by the reviewer, the title has been revised and rewritten. (Lines 1-5).

(ii) Introduction: Huge modifications are necessary for the introduction section.

The research gaps and significance of the study are not shown.

Moreover, the authors should provide more precise research objectives in this study.

Also, please highlight some incidents of heavy metal pollution that occurred in the region which drove you to perform the study.

As suggested by the reviewer, the significance of the study has been discussed thoroughly in the introduction (see Lines 37-51). Furthermore, the gap and the research objectives are highlighted in the introduction as well (see Lines 61-65).

As for incidents of heavy metal pollution that occurred in the regions, the authors have clearly demonstrated the importance of continuous water quality monitoring in the area, whether or not previous incidents have occurred.  

(iii) The reasons spatial distribution of the heavy metals were not clearly illustrated, for instance why a certain area has higher pollution than other places?

(iv) Discuss the upstream and downstream differences in this study. I would suggest to incorporate the landuse type into the discussion.

There are not upstream and downstream in the study area. Samples have been collected from the surface seawaters along the Red Sea coast and not a river basin.

(v).  So what are the proposed mitigations? What should the government do for proper water quality management? Effects from changing land use or anthropogenic activities? The difference between upstream or downstream is not provided.

(vi).  Some relevant literature is missing in this study.

a.      doi:10.1007/s10661-020-08543-4.

b.      doi: 10.1007/s10661-021-09202-y

(vii) I would suggest replacing Table 2 with a box plot for better illustration. Also, for the MAC, it is following the WHO values? Do you have your own standards/limits in Saudi Arabia?

(viii) Results in section 3.2 seems to be a very general discussion, and not region specific. I cannot follow the main reasons for the causes. Please revise.

(ix) Conclusion section seems to be a repetition of the results section. Huge modifications are required. Please provide insights into this study and what can be further done in the future.

Conclusion modified

(x) I do not find the significance of determining the correlation between parameters in this study. Please provide more details. I understand that you would like to include different statistical analyses for enriching the study, however, the flow is not there. Please revise.

Round 2

Reviewer 2 Report

This paper is an interesting study and authors have investigated the potential environmental impact and the possible sources of heavy metals (HMs), and evaluated the quality of coastal surface seawater in the vicinity of Yanbu, along the Red Sea coast of Saudi Arabia. Because water environment protections are one of the most important problems, it is necessary to find new methods and techniques to monitor point sources of pollution in the Red Sea water.

The article is written correctly, includes a discussion of the research findings, and a good review of the literature. The results are presented in a clearly structured manner. The manuscript has been significantly improved and can now be accepted in current form.

Author Response

Manuscript Number: water-2090012

Full Title: Water quality and ecological risk assessment of heavy metals in the coastal seawater of the Red Sea, Saudi Arabia

This paper is an interesting study and authors have investigated the potential environmental impact and the possible sources of heavy metals (HMs), and evaluated the quality of coastal surface seawater in the vicinity of Yanbu, along the Red Sea coast of Saudi Arabia. Because water environment protections are one of the most important problems, it is necessary to find new methods and techniques to monitor point sources of pollution in the Red Sea water.

The article is written correctly, includes a discussion of the research findings, and a good review of the literature. The results are presented in a clearly structured manner. The manuscript has been significantly improved and can now be accepted in current form.

Kind regards,

We would like to thank you for your valuable comments which improve our manuscript.

Reviewer 3 Report

The authors have substantially addressed my comments.

Author Response

The authors have substantially addressed my comments.

Thanks for your valuable comments which improved our manuscript.
